# Spectroscopic Evidence for Photooxidation of Tocopherols in n-Hexane

**DOI:** 10.3390/molecules26030571

**Published:** 2021-01-22

**Authors:** Bogdan Smyk

**Affiliations:** Department of Physics and Biophysics, The Faculty of Food Sciences, University of Warmia and Mazury in Olsztyn, Oczapowskiego 4, 10-719 Olsztyn, Poland; bsmyk@uwm.edu.pl; Tel.: +48-89-5233556

**Keywords:** tocopherols, photooxidation, absorption, fluorescence, time-resolved, lifetimes, C-T complexes

## Abstract

This paper presents the results of an investigation into the photooxidation of tocopherols (Tocs) dissolved in argonated and non-argonated n-hexane. During irradiation, steady-state absorption and fluorescence spectra as well as lifetimes were measured. In all experiments, the photoreactions were of the first order type. The reaction rate was higher for all Tocs in argonated solvent. A new emission band with a maximum at 298 nm as well as new absorption and fluorescence bands beyond the 300 nm connected with charge-transfer (C-T) complexes for all Tocs appeared during the irradiation of γ- and δ-Toc. The above results indicate that the photooxidation process is very complex and that the observed phenomena strongly depend on the number and position of methyl groups in the chromanol ring.

## 1. Introduction

Tocopherols (Tocs) have been known from almost 100 years and addressed in many in vivo and in vitro experiments as a very important vitamin E which has to be supplied with food. Tocs are naturally present in different parts of plants. In the photosynthetic apparatus they can be found in bound form in cell membranes. Their most important function is to protect lipids in the membrane from oxidation mainly by scavenging reactive oxygen species and breaking the chain-propagation reactions of unsaturated fatty acids [1,2]. This is feasible because of the occurrence of a -OH group in the chromanol ring of Tocs (Appendix A) from which the H atom can be abstracted. One electron oxidation results in the formation of tocopheroxyl radical [3], which is inactive and does not propagate lipid oxidation. Two radicals can form a dimer or dihydroxy dimer having a planar structure [4,5,6,7,8]. Higher complexes—trimers—were also observed [4,5,7,8,9,10,11]. One-electron oxidation can be followed by two-electron oxidation, which can result in the formation of tocopherylquinone (TQ) or spirodimer [6,7,12]. It seems that these molecules are the final products of Tocs oxidation. Tocs together with carotenes play the role of lipophilic antioxidants in oils. They can scavenge singlet oxygen by physical action [13,14,15,16] and chemical reactions [13,17,18], revealing a synergistic action [19,20] which, however, has not been fully elucidated. Moreover, in my previous investigation in the model system, α-Toc was the worst scavenger of all the Tocs analyzed [21]. Tocs exist in different racemic forms, the quantity of which depends on the number of methyl groups in a phytol chain (Appendix A). Because of the presence of three methyl groups, the number of racemic forms is eight [22]. The most commonly studied Tocs are α-Toc and γ-Toc, due to their physiological effects on human and animal organisms [6,7,22,23,24,25]. α-Toc has also been implicated in the prevention of UVB induced skin photodamage [7,8] due to the absorption band lying in this area. Many investigations were performed concerning the oxidation of Tocs [2,4,19,20,26,27] but the number of works addressing their photooxidation is not that high [5,6,8,28,29,30]. Therefore, the goal of this study was to investigate UVB radiation-induced photooxidation of all Tocs in n-hexane under aerobic and anaerobic conditions using steady state and time resolved spectroscopic methods.

## 2. Results and Discussion

### 2.1. Absorption Spectra

Tocs were irradiated with lamps equipped with UV filters: mainly through UV1 (FWHM: 280–300 nm) and UG11 (Schott filter, FWHM: 270–380 nm). Their normalized absorption and transmission spectra are shown in Figure 1a. Transmission of both filters very well overlapped with the absorption bands of Tocs. The absorption spectra of α-Toc during irradiation are shown in Figure 1b, those of β-Toc in Appendix A, those of γ-Toc in Appendix A, and those of δ-Toc in Appendix A. In each figure, the blue line depicts the initial non-irradiated spectrum and the red line the final one. The first spectrum was measured after 5 min of irradiation, which continued up to 60–80 min. We can observe different behavior, probably because of different structures of chromanol rings. Their chemical structure differs mainly in the methyl group location in position 5 in α- and β-Toc and in the lack of this group in δ- and γ-Toc (Appendix A). Two isosbestic points are created on both sides of the main band; they suggest an existing equilibrium between different forms of Tocs molecules. The left-hand side isosbestic points at 274 nm are due to the equilibrium between oxidized and unoxidized forms of Tocs. The oxidized forms can be different and depend on the oxidation of one or two electrons. There are many reports on the oxidized forms existing in different systems, and it seems that the main one is tocopherylquinone (α-TQ) formed upon the two-electrons oxidation [26] that has an absorption maximum at approximately 260–268 nm [31]. It can also be seen in Appendix A. The second form can be tocopheroxide with a maximum at 240 nm [31] or a non-reactive radical, both formed due to the one-electron oxidation, further called the “first product” (FP). To find the absorption spectra of the oxidation products, the absorption of each Toc before and after irradiation was decomposed into Gaussian bands and additionally the absorption spectrum of α-TQ (commercially bought) was measured with and without α-Toc (Appendix A). In this figure, the black line depicts α-TQ and the blue one depicts α-TQ + α-Toc spectrum. It seems that the α-TQ spectrum is a good candidate for a photooxidation product, although the characteristic two peaks with the maximum of approximately instead 260–268 nm are not seen in Figure 1b. These maxima can be seen in Appendix A at approximately 274 and 284 nm instead. They can correspond to δ-TQ because the main band of δ-Toc is bathochromically shifted compared to α-Toc. However, the main photoproduct seems to have its maximum at approximately instead 240 nm (Figure 1b) and its spectrum does not overlap with that of Tocs—therefore, has no influence on the left isosbestic point (Appendix A). The analysis revealed that peaks with maxima at 285 and 257 nm disappeared (Appendix A) and new ones with maxima at 268 and 240 nm appeared during irradiation (Appendix A). This suggests that the α-Toc band consists of two or three bands of racemic forms. One of them disappeared faster and two new bands appeared that were related to products of photooxidation. The band with maximum at 268 nm can correspond to α-TQ and the second one with the maximum at 240 nm—to FP. The same analysis was done with other Tocs and results for δ-Toc are shown in Appendix A. The peak at 278 nm (Appendix A) can probably correspond to δ-tocoferylquinone (δ-TQ) and the peak at 239 nm to FP. Similar peaks were found for β-Toc (271 and 236 nm) and for γ-Toc (274 and 243 nm). A second isosbestic point presents for α-Toc (Figure 1b) at 307 nm on the right side of the band and similar points are shown in Appendix A for the other Tocs. These points suggest a second equilibrium between Tocs and photoproducts which might be due to the charge-transfer (C-T) complexes formed. The new bands are again similar for the pairs of: α- and β-Toc (Figure 1b, Appendix A), and γ- and δ-Toc (Appendix A). It is well known that C-T complexes are formed between Tocs and quinones [32,33,34]. Quinones are electron acceptors and Tocs are electron donors; this should allow for forming stacking complexes (the sandwich structure). It is also known that generally not only one type of such a dimer structure could exist, but the “isomeric” forms as well [33], having maxima at different wavelengths. In addition, Figure 1b and Appendix A show a few maxima above 300 nm which indicate that a few complexes can be formed by every Toc. The question is whether FP can also form such C-T complexes. To check the formation of C-T complexes, an experiment was performed with the UG11 filter, where C-T complexes have absorption bands (Figure 1a,b). The number of photons transmitted by UG11 was higher compared to the number determined for the UV1 filter. Therefore, the reaction rate should be higher for this filter, and absorbance above 320 nm should also be much higher. Comparing absorbances in Appendix A, it can be seen that the reaction is at almost the same stage for both filters (red lines). Initial absorbances at the absorption maximum in both spectra are also almost the same. Therefore, it is possible to compare absorbance ratios at 297 nm to 340 nm, which will show the influence of the photons transmitted through the UG11 filter. This relation yielded the value of 4.7 for the UV1 filter and the value of 8.8 for the UG11 filter. The ratios for the UV1 filter were much lower (about two times) than for the UG11 filter, which indicates that photons in the range of 300–450 nm have enough energy to destroy these complexes, which further indicates that the bonding energy is not very high.

### 2.2. Rate Constants from Absorption Spectra

Reaction rate constants were calculated for the region 280–300 nm (Tocs) using one exponential decay function (Abs (t) = Span*exp(−K*t) + Plateau), and for the regions: 240–270 nm (products) and 320–400 nm (C-T complexes) using one exponential association function (Abs (t) = Abs(0) + (Plateau − Abs(0))*(1 − exp(−K*t))) with 1 nm step starting with the initial spectrum (when it was possible) and ending with one of the last ones. In each wavelength range, it was not possible to separate a sub-region in which the rate constants would have values oscillating around the mean (Appendix A). Such a situation takes place because of the overlapping bands of Tocs, products, and C-T complexes and because of the appearance and disappearance of other bands. For these reasons, the rate constants were not shown. Moreover, for δ-Toc at the first stage of reaction, the absorbance slightly increases or remains constant up to about 200 min This causes great errors in reaction rate calculations, which is shown in Appendix A. This can be explained by the appearance of a new band with a maximum at 278 nm (Appendix A), the increase of which counteracts the decrease in absorbance at the maximum of the δ-Toc band. Results presented above, connected with absorption spectra are difficult to interpret. Discussion would be easier if the Tocs were forming stacking dimers, trimers, or clathrates. Only a few reports are available in the literature, but there are no dimers of the stacking structure but rather the co-planar one [7,8,9,35] and solution concentrations are more than two orders of magnitude higher than those used in this study [35]. Moreover, concentration dependence of absorbance (Lambert-Beer law checking—data not shown) gave strictly straight lines with R^2^ of about 0.9999, which indicates that no dimers were formed. However, if one could make such an assumption, the explanation of some of the results could be feasible. The 3D structure [36] of α-Toc suggests that the spatial conformation does not exclude the formation of stacking dimers. Dimers could be formed more easily between Tocs having one or two methyl groups than three due to a steric barrier. Presented data suggest that the observed phenomena depended on the number and position of the methyl group. Also, C-T complex formation would be easier to explain because the reaction would not be diffusion-controlled. Only one Toc molecule will be oxidized remaining in the dimer and forming a C-T complex. The next interesting behaviour found for all Tocs during irradiation was higher velocity—higher rate constants observed for argonated samples. The effect of deoxygenation is seen in Figure 2, velocities of α-, and β-Toc are almost the same and different from those of δ- and γ-Toc. These data strongly suggest that the number and position of -CH_3_ group in the chromanol ring play a crucial role, but the higher rate of photooxidation at the lower oxygen concentration is unexpected. The first step of the photooxidation reaction can be the hydrogen atom abstraction from the hydroxyl group, causing the formation of the tocopheroxyl radical, and simultaneously, the oxygen superoxide which, according to the reaction:TocO + O_2_^−^ + H^+^ → TocOH + O_2_(1)
partially reforms the initial concentration of the Toc molecule, which may explain the higher reaction rate of argonated samples. The reasons for this could also be the lower microviscosity of the samples (not measured) for diffusion-controlled reactions or the fact that the reaction was not diffusion-controlled, i.e., oxygen could participate in Tocs adducts formation. This is a known phenomenon (oxygen bridges), but rather as the last stage—termination stage of oxygenation chain reaction of unsaturated fatty acids in plant oils [2]. Argon presence could disturb the formation of such bridges. Whether photooxidation reactions are diffusion-controlled or not is still an open question and further experiments are needed in this respect.

### 2.3. Fluorescence

After absorption spectrum measurements, three fluorescence spectra were collected practically at the same time: two emission spectra and one excitation spectrum with a high speed of registration. When irradiation was stopped, photooxygenation reaction was also arrested. Measurements of four spectra and decay time for one wavelength of observation lasted about 4 min. Responses of the samples differed for the two above-mentioned pairs of Tocs. Figure 3 presents data for α-Toc and δ-Toc. Spectral changes of β-Toc were similar to those of α- and to a lesser extent to those of γ- and δ-Toc. In the Figure 3, changes are parallel for absorption for α-Toc and not parallel for those δ-Toc. Firstly, there is no “incubation” period and, secondly, a new band appears with the maximum at approx. 298 nm. This new band cannot correspond to a quinone-like structure, because quinones do not fluoresce [37] due to the lack of conjugated double bonds. Indeed, α-TQ excitation gave zero fluorescence, which was checked in separate experiments. It seems that fluorescence spectra: emission and excitation, are better than the absorption ones to calculate the rate constants because the fluorescence spectra are homogeneous except for γ- and δ-Toc where new products band overlap partially with the band of γ- and δ-Toc. Moreover, the intensity changes do not indicate any “incubation” period like the data from absorption measurements. The rate constants presented in Table 1 were calculated based on the area under the fluorescence spectrum for λ_exc_ = 265 and 283 nm. For γ- and δ-Toc, the area under the spectrum was taken in the range of 310–400 nm due to the spectral overlap. For the excitation spectra, the same procedure was used as for the emission spectra but in the range from 250 to 320 nm. The values of rate constants have higher velocity of the argonated samples. The new band with the maximum at 298 nm was growing faster also for the argonated sample, which is seen in Appendix A. Kinetics connected with this new band differed from all the others and did not depend on the method of calculation used: it holds for any wavelength in the band range or area under the spectrum in the range where the spectra of new band and δ-Toc do not overlap. This kinetics has a sigmoid-like shape (Appendix A). It can be seen that the growth rate is higher for the argonated sample, which confirms the higher reaction rate of this sample.

### 2.4. Fluorescence of C-T Complexes

C-T complexes were formed between unoxidized and oxidized molecules of Tocs. Excitation attempts of these complexes were successful, although fluorescence was very weak, and measurements have been done with twice wider spectrofluorometer slits: 10 and 10 nm. It means that the registered signal was about 16 times weaker than for 5 and 5 nm slits. However, it was possible to excite the C-T complexes. Wavelengths of excitation were from 340 to 390 nm and observation was made from 400 to 430 nm depending on the Toc. Emission spectra of α-Toc are shown in Appendix A and these of δ-Toc in Appendix A, whereas excitation spectra are presented in Appendix A for α- and δ-Toc, respectively. Spectra in Appendix A show two maxima, one at approx. 380 nm and the other at 400 nm, which indicates that probably two kinds of C-T complexes exist. Their absorption maxima are poorly noticeable above 320 nm in the excitation spectra in Appendix A. The maximum at approx. 290 nm in this figure corresponds to α-Toc. A different situation is observed in Appendix A which presents emission spectra of δ-Toc C-T complexes. Fluorescence intensity of these complexes is about four times higher than that of α-Toc despite the lower absorbance at excitation wavelength (Appendix A), which points to a higher quantum yield. Two different maxima are also present: one at 428 nm (λ_exc_ = 360 nm) and the other at 443 nm (λ_exc_ = 390 nm)—at least two different complexes exist as indicated by maxima of the excitation spectra in Appendix A. The first absorption maximum was located at a wavelength of 362 nm (λ_obs_ = 430 nm) and the second occurred at 380 nm (λ_obs_ = 490 nm). These maxima are not shown in Appendix A. In turn, Appendix A reveals a maximum at approx. 267 nm, which cannot correspond to either δ-Toc or to δ-TQ. This absorption band corresponds to a new emission band of a product of δ-Toc which has its maximum at 298 nm (Figure 3d,e and Appendix A). This band cannot be seen in Figure 3f due to low intensity. Figure 3d,e indicate that this fluorescence maximum at 298 nm is better excited by 265 nm than by 283 nm wavelength, because of smaller absorbance at an excitation wavelength of 265 nm and almost the same intensity of both, which confirms the above results. This band is missing in Appendix A for α-Toc, as such a new band like that for δ-Toc was not observed. C-T complexes formed for β- and γ-Toc were similar to those formed for α- and δ-Toc but their spectra were not shown. The existence of the C-T complexes requires additional confirmation. It seems that the new method proposed by Wang et al. [38] or a plasmonic fluorescence method could be used to enhance their very low fluorescence intensity.

### 2.5. Fluorescence Lifetimes

Fluorescence decay times for excitation wavelength at 283 nm and emission wavelength at 325 nm were registered after measuring absorption and fluorescence spectra during time-course experiments. Additional separate series of measurements for non-argonated samples were done using time resolved emission spectra (TRES) for 283 nm excitation wavelength and observation wavelength from 300 to 370 nm with 5 nm step. The Global method was used to develop the results in both series. In the first case, changes of f_i_ coefficients connected with the reconvolution method (Appendix A) were presented. For all Tocs, the reaction rate was higher for the argonated samples, which confirmed data obtained from absorption and fluorescence measurements. This figure shows that bi-exponential decay was observed only in the case of α-Toc in contrast to the other Tocs where tri-exponential decay occurred. The long-lived component was present in all Tocs, but not in the samples at the beginning of the experiment. This is especially evident for γ-Toc (Appendix A) and δ-Toc (Appendix A). The long-lived component of all Tocs slightly differed and its steady-state spectra could be seen only for the γ- (data not shown) and δ-Toc (Figure 3d,e). For α- and β-Toc, only traces were registered as a new band, which indicates that their intensity was very low or was interfered by the main band. Moreover, all lifetimes were longer for deoxygenated samples, which should be attributed to fluorescence quenching by oxygen. Quenching should shorten lifetimes especially for long living components because the time for collisional quenching in the excited state is longer and thus the probability is higher. Therefore, they are shortened much more if lifetimes are longer. From the presented data it is not obvious what components the lifetimes should relate to. For this purpose, TRES data of samples at the end of irradiation presented in Figure 4 were analyzed. Again, it is seen that the spectra calculated with 0.5 ns step in pairs are similar: α- and β- as well as γ- and δ-Toc. Components with long lifetimes (red lines) have in each case emission spectra shifted toward shorter wavelengths compared to the main band. Therefore, the longest lifetime should relate to the new fluorescence spectrum with maximum below 300 nm. However, it remains unknown which steady-state spectrum of the two components with lifetime from 1.3 to 1.4 ns and from 0.2 to 0.4 ns should be combined with. It seems that a better answer is provided by the spectra obtained by multiplication of f_i_ by coefficients obtained from TRES calculation by the spectrum intensity I(λ) (decay-associated spectra (DAS)) (Appendix A—DAS spectra). Green and red spectra can be rather connected with the non-oxidized form of Tocs poorly separated among themselves and the long lifetime component (blue line) with the spectrum appearing because of photooxidation.

## 3. Materials and Methods

### 3.1. Materials

Tocopherols: (±)-α-tocopherol (α-Toc) (98.6%), rac-β-tocopherol solution (β-Toc) (GC > 99%, HPLC > 98%), (+)-δ-tocopherol (δ-Toc) (96%), (+)-γ-tocopherol (γ-Toc) (97%), and D-α-tocopherylquinone (α-TQ), were from Sigma-Aldrich (Poland) and were used without further purification. Extinction coefficients in n-hexane were calculated as average values based on some non-irradiated and non-deoxygenated samples in different experiments. For particular Tocs, they were: α-Toc (297 nm) = 3596 ± 46 mol^−1^ L cm^−1^, β-Toc (298 nm) = 3740 mol^−1^ L cm^−1^—only one sample, δ-Toc (295 nm) = 3540 ± 40 mol^−1^ L cm^−1^, and γ-Toc (295 nm) = 4110 ± 110 mol^−1^ L cm^−1^. Concentration of each tocopherol sample was chosen so that the absorbance was in the range between 0.4 and 1.1. n-Hexane was from Merck (Poland) and argon 5.0 (99.999%) was from Eurogaz-Bombi (Poland).

### 3.2. Methods

#### 3.2.1. Deoxygenation

n-Hexane was poured into a 20-mL clear glass vial with a metal screw headspace (Alwsci Technologies, Shaoxing, China), with a Teflon seal instead of the original seal, with two holes. One hole was empty, whereas a Pasteur pipette was inserted into to the second one. The pipette was connected with the tube and an argon container. Bubbling lasted four hours with a very slow flow. After completed bubbling, the sample was quickly dissolved in deoxygenated n-hexane on a spatula and transferred to a fluorescence 1 × 1 cm quartz cuvette with a plug.

#### 3.2.2. Samples Irradiation

Samples were irradiated using an XBO 150 W/CR OFR lamp (Osram GmbH, Munich, Germany) working in a horizontal position. The distance between the lamp (in the housing) and the sample in the horizontal position cuvette was approximately 27 cm. Between them there were: water filter to avoid thermal effects, and absorption wide band filters: UV1 (FWHM: 280–300 nm)—prepared by Optel (Opole, Poland), and UG11 (Schott filter, FWHM: 270–380 nm) bought from Edmund Optics (Finland). Their normalized absorption and transmission spectra are shown in Figure 1a. The power of the radiation in the sample position was measured using an R-752 Universal Laser Radiometer with PH-30 Power Head (Digi Rad, Palo Alto, CA, USA). In order to exclude the effect of the long-wave radiation transmitted by the above filters on the total measured power density (Figure 1a—inset), these filters were assembled with a cut-off 600 nm filter. This allowed estimating density of radiation power in the UV range. The power density transmitted by both UV filters after this correction was: 0.2 mW cm^−2^ for UV1 and 6.0 mW cm^−2^ for UG11. Irradiation above 650 nm transmitted by the filters had no effect on the Tocs. In this set-up, the samples were irradiated for 5–80 min depending on the stage of the process. Taking out the sample stopped the process, and then measurements could be done.

#### 3.2.3. Apparatus

Absorption spectra were measured using a Cary 5000 (Agilent, Mulgrave, VIC, Australia) spectrometer, and fluorescence spectra using a Cary Eclipse (Agilent, Australia) fluorimeter in the right-angle geometry. Both instruments were equipped with a Peltier accessory. All measurements were carried out in a tightly closed 1 × 1 cm fluorescence quartz cell. Excitation and emission slits were set at 5 nm and for charge-transfer (C-T) complexes both at 10 nm. Temperature was stabilized at 24 °C. Samples of each Toc were excited at 265 nm and 283 nm, and then observation was made at 325 nm for excitation spectra. C-T complexes were excited at 330 to 390 nm and observations were made at 420 to 490 nm. Measurements were done at different PMT voltages. Correction to 600 V (default voltage) has been done for each voltage using experimentally obtained coefficients. Every emission and excitation spectrum was corrected for wavelength-dependent instrument sensitivity curve and inner filter effects I and II [40]. Rayleigh and Raman scatterings were measured using blank samples and were subtracted from each emission and excitation spectrum using Grams Ai Spectroscopy Software (Thermo Fisher Scientific, Waltham, MA, USA). All other calculations were done using GraphPad Prism ver. 7.05. (GraphPad Software Inc., San Diego, CA, USA).

Fluorescence lifetimes were measured using a FluoTime 200 spectrometer (PicoQuant, Berlin, Germany) with a TCSPC module and an MCP PMT detector. The Led Diode 283 nm (PicoQuant, Germany) served as the source of excitation. A cell holder with the right-angle geometry and 4 ps resolution was used. The count rate per second at the detector was kept below 1% of the laser replication rate to avoid pulse-pileup. The excitation wavelength was set at 283 nm and observation wavelength at 325 nm in one type of experiments, and from 300 to 370 nm to measure the time resolved emission spectra (TRES). Data was analyzed with FluoFit 4.6.6 version software (PicoQuant, Germany) using the multiexponential intensity decay model as follows:(2)$$I\left(t\right)=\overset{t}{\underset{-\infty }{\int }}IRF\;(t^{\prime })\overset{n}{\underset{i=1}{\sum }}\alpha_{i}e^{-\frac{t-t^{\prime }}{\tau_{i}}}dt^{\prime }$$
where IRF (*t′*) is the instrument response function at time *t′*, *α_i_* is the amplitude of the decay of the i-th component at time *t*, and *τ_i_* is the lifetime of the *i*-th component. Goodness of fit was estimated by calculating χ^2^, using plane error analysis (PEA) and autocorrelation function implemented in FluoFit program. Lifetimes were calculated using the reconvolution model and the Global method for time resolved emission spectra (TRES) as well as for one wavelength measurements. TRES spectra were calculated using methods provided by Maroncelli & Fleming [39].

## 4. Conclusions

The presented data are difficult to interpret. One of the possibilities is dimer or trimer formation but not through covalent bonds, but the checking of dimer formation in the investigated concentration range gave negative results. This is an attractive hypothesis, especially in the context of C-T complexes formed by oxidized and unoxidized forms of Tocs, which appear on the long wavelength side of the main band. Molecules in such complexes must be close together and rather in stacking conformation. The second observed phenomenon relates to argonated samples for which higher reaction rate was observed as opposed to the non-argonated ones. It suggests that oxygen takes part in the stabilization of Tocs of unoxidized molecules (self-organization?) or the argon can reduce microviscosity of the solution. However, for α- and β-Toc, the difference between rate constants was not very big and occurred mainly for γ- and δ-Toc. The next phenomenon was related to a new emission band appearing for γ- and δ-Toc during photooxidation without the action of any enzyme. The moiety possessing this new band has much longer lifetime and higher quantum yield than γ- or δ-Toc, which suggests greater rigidity of this molecule. The maximum of the absorption band of this moiety, obtained from excitation spectra, is located at about 270 nm. All of the above results indicate that the observed phenomena depend on the number and position of the methyl groups in the chromanol ring. The question is whether the dimer of Tocs can exist in membranes of photosynthetic units in the bound form or in the free form as, e.g., in vegetable oils.

## Figures and Tables

**Figure 1 molecules-26-00571-f001:**
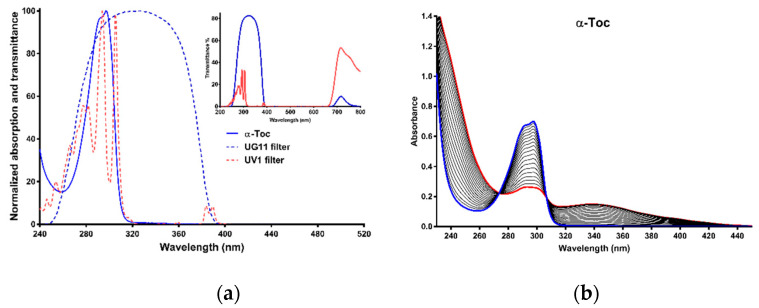
(**a**) Normalized absorption spectra of α-Toc and normalized transmittance spectra of band pass filter used. Inset—transmittance spectra of filters in full UV+Vis range. (**b**) Time-course of absorption spectra of α-Toc. Blue line—initial spectrum, red line—final spectrum. Irradiation time 421 min, optical path—1 cm.

**Figure 2 molecules-26-00571-f002:**
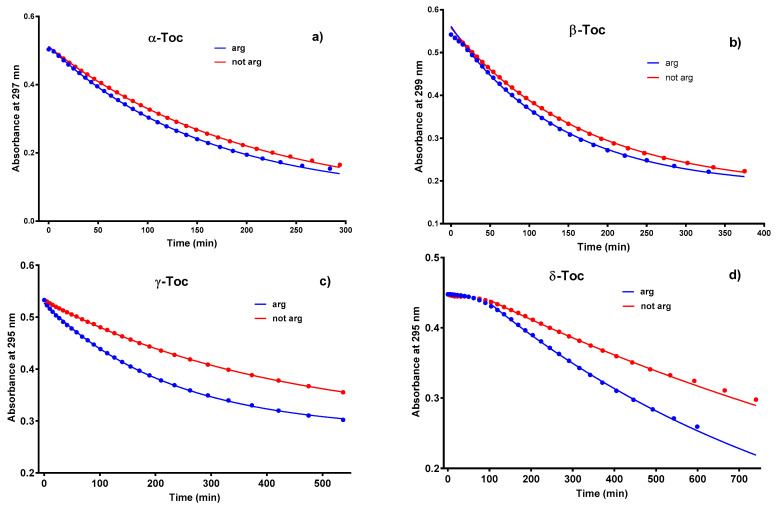
Time-course of absorbance at maximum of Tocs samples: argonated during irradiation—blue lines and non-argonated—red lines. One exponential decay function was used to fit experimental data.

**Figure 3 molecules-26-00571-f003:**
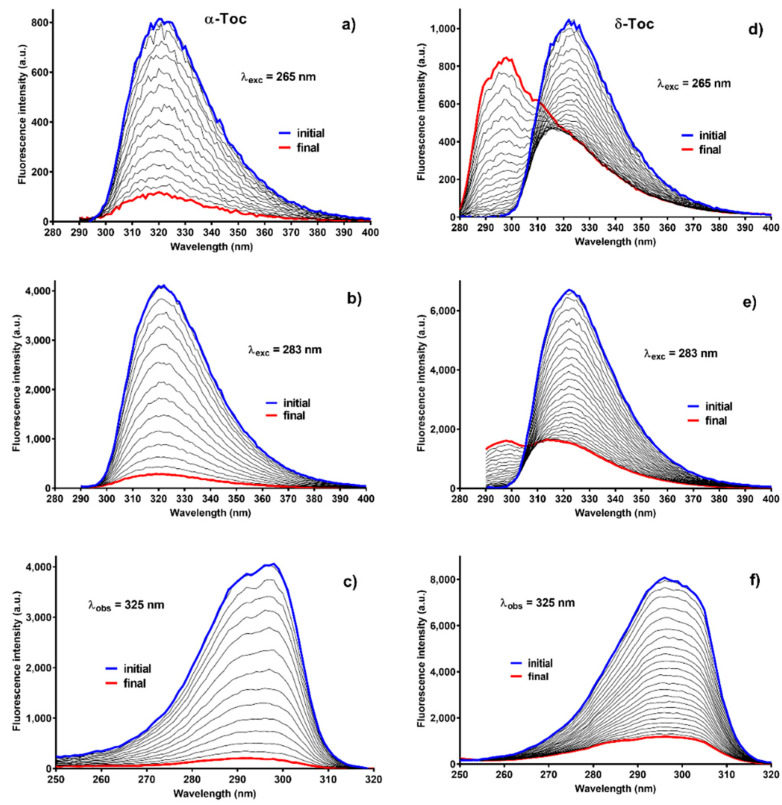
Time-course of Tocs fluorescence spectra: (**a**–**c**)—α-Toc, (**d**–**f**)—δ-Toc. (**a**,**b**,**d**,**e**)—emission spectra, (**c**,**f**)—excitation spectra. Irradiation time: α-Toc—409 min, δ-Toc—905 min. λ_obs_ = 325 nm, λ_ex_ = 265 and 283 nm.

**Figure 4 molecules-26-00571-f004:**
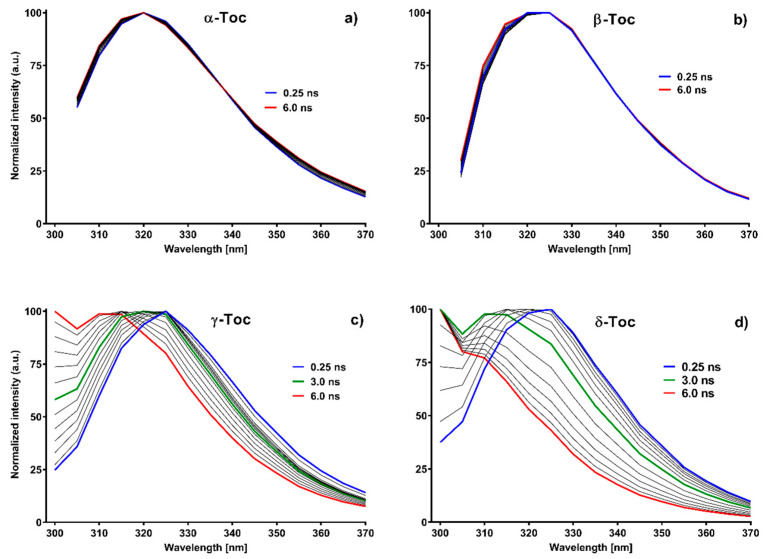
Calculated 2D TRES spectra at the end of irradiation of Tocs. Time step—0.5 ns. Blue lines initial spectra, red lines—final ones. Green spectra are middle spectra. Spectra were calculated using Global method and [39].

**Table 1 molecules-26-00571-t001:** Photooxidation rate constants of tocopherols in n-hexane.

		Fluorescence	
		λ_exc_ = 283 nm k × 10^3^ [min^−1^]	λ_obs_ = 325 nm k × 10^3^ [min^−1^]
**α-Toc**	arg	8.86 ± 0.52	9.31 ± 0.57
	non-arg	5.89 ± 0.47	6.20 ± 0.73
**β-Toc**	arg	5.69 ± 0.18	6.20 ± 0.14
	non-arg	4.63 ± 0.11	4.80 ± 0.11
**γ-Toc**	arg	13.10 ± 0.15	12.18 ± 0.14
	non-arg	7.21 ± 0.84	7.38 ± 0.18
**δ-Toc**	arg	9.30 ± 0.15	7.73 ± 0.05
	non-arg	3.36 ± 0.32	1.27 ± 0.17

Error = SD—standard deviation. Fluorescence rate constants for all Tocs were calculated using an area under spectra in the range from 305 to 400 nm for emission spectrum and from 250 to 320 nm for excitation spectrum.

## Data Availability

Data is contained within the article or supplementary material.

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
