# Peer review of "Spectroscopic Evidence for Photooxidation of Tocopherols in n-Hexane"

_molecules, 2021, doi:10.3390/molecules26030571_

Round 1

Reviewer 1 Report

The author has taken my advice and greatly improved the manuscript by removing the part dealing with the rate coefficients derived from the absorption maxima, which I am grateful for. 

The only major thing that I think still would need some elaboration is what the author states in Section 2.3:

"The values of rate constants confirmed data obtained from absorption analyses: concentration dependence and higher velocity of the arrogated samples."

Does this sentence also mean that there are multiple rate constants derived from the fluorescence spectra at different concentration values? If so, the data for different concentrations should be added to the manuscript (or to the Supplementary Material). Since overlapping is not that much of an issue here than for the absorption spectra, it would definitely confirm the conclusions drawn in Section 2.2 (i.e. that dimers form in the sample resulting in the observed concentration dependence).

A couple of other, minor issues I found in the manuscript – probably there are more than the ones listed below:

  1. Line 15, abstract: "a new emission band with a maximum at 298 nm appeared together with ... and absorption and fluorescence bands connected with charge-transfer (C-T) complexes." – something is missing from the sentence
  2. Line 67: "The left-hand side isosbestic points at 274 nm are created despite oxidized and unoxidized forms." – "between" would be probably better than "despite"
  3. Line 73: "first product (FP).To find the absorption spectra" – missing space between the sentences
  4. There is no Figure 2 in the manuscript.
  5. Line 102: "Chart A1" – there is no 'Chart A1' only 'Chart S1' but I do not see the connection here. Maybe the author actually meant Figure 1a instead?
  6. Line 139: "methyl groups then three dues to a steric barrier" – 'than' instead of 'then', 'due to' instead of 'dues to'
  7. Line 140: "Also, C-T complexes formation" – use the singular form of 'complex' here
  8. Line 162: "Spectra changes" to "Spectral changes"
  9. Line 173, Table 1: "k x 10ˆ3 [min-1]" – I would suppose that there is a negative sign missing here, i.e. "k x 10ˆ-3" maybe?
  10. Line 231: "This Figure hows" to "This Figure shows"

Author Response

Reviewer 1

The author has taken my advice and greatly improved the manuscript by removing the part dealing with the rate coefficients derived from the absorption maxima, which I am grateful for. 

The only major thing that I think still would need some elaboration is what the author states in Section 2.3:

"The values of rate constants confirmed data obtained from absorption analyses: concentration dependence and higher velocity of the arrogated samples."

This sentence was removed.

Does this sentence also mean that there are multiple rate constants derived from the fluorescence spectra at different concentration values? If so, the data for different concentrations should be added to the manuscript (or to the Supplementary Material). Since overlapping is not that much of an issue here than for the absorption spectra, it would definitely confirm the conclusions drawn in Section 2.2 (i.e. that dimers form in the sample resulting in the observed concentration dependence).

Almost all sentences in each part of manuscript connected with concentration dependence of rate constants have been removed.

A couple of other, minor issues I found in the manuscript – probably there are more than the ones listed below:

  1. Line 15, abstract: "a new emission band with a maximum at 298 nm appeared together with ... and absorption and fluorescence bands connected with charge-transfer (C-T) complexes." – something is missing from the sentence

A new emission band with a maximum at 298 nm as well as new absorption and fluorescence bands beyond the 300 nm connected with charge-transfer (C-T) complexes for all Tocs appeared during irradiation of γ- and δ-Toc.

  1. Line 67: "The left-hand side isosbestic points at 274 nm are created despite oxidized and unoxidized forms." – "between" would be probably better than "despite"
  2. Line 73: "first product (FP).To find the absorption spectra" – missing space between the sentences
  3. There is no Figure 2 in the manuscript.
  4. Line 102: "Chart A1" ­– there is no 'Chart A1' only 'Chart S1' but I do not see the connection here. Maybe the author actually meant Figure 1a instead?
  5. Line 139: "methyl groups then three dues to a steric barrier" – 'than' instead of 'then', 'due to' instead of 'dues to'
  6. Line 140: "Also, C-T complexes formation" – use the singular form of 'complex' here
  7. Line 162: "Spectra changes" to "Spectral changes"
  8. Line 173, Table 1: "k x 10ˆ3 [min-1]" – I would suppose that there is a negative sign missing here, i.e. "k x 10ˆ-3" maybe?
  9. Line 231: "This Figure hows" to "This Figure shows"

All errors have been corrected and the figure numbers have been renumerated.

Reviewer 2 Report

REVIEW OF REVISED MANUSCRIPT “SPECTROSCOPIC EVIDENCE…”, BY SMYK.

The author has made a good job while revising the manuscript. Written in better form, even the part I was most sceptical about now looks reasonably convincing. I have still some suggestions for improvement, among which there is also a hint about the possible explanation of the oxygen effect on degradation.

  1. Line 9. “into on the” should read “into the”.
  2. Line 15. “together with and” should read “together with”.
  3. Line 27. “oxidation mainly” should read “oxidation, mainly by”.
  4. Line 29. Delete “presence”.
  5. Line 30. “result” should read “results”.
  6. Line 32. “Oxidation of one electron” should read “One-electron oxidation”.
  7. Line 33. “by second” should read “by a second”.
  8. Line 35. “scavenge a singlet” should read “scavenge singlet”.
  9. Line 37. “fully yet” should read “fully elucidated”.
  10. Line 50. “with UV” should read “with lamps equipped with UV”.
  11. Line 66. “are created” should read “are due to the”.
  12. Line 67. Delete “despite”.
  13. Line 68. “oxidation” should read “abstraction” (also in lines 70 & 72).
  14. Line 70. “that have” should read “that has”.
  15. Line 91. “created for…points in” should read “present for…points are shown in”.
  16. Line 93. “and that” should read “, which”.
  17. Line 99. “these Figures”: which ones? Please specify. “beyond” should read “above” (also in line 104).
  18. Line 100. “form such…check the” should read “form such C-T…check for the”.
  19. Line 101. Delete “transmitted photons”.
  20. Line 103. “higher…UV1” should read “higher compared to the UV1”.
  21. Line 108. “through UG11” should read “through the UG11”.
  22. Line 120. “rate decreased” should read “rate constant decreased”.
  23. Line 121. “reaction for the” should read “reaction for an”.
  24. Line 122. “exponent…when for” should read “exponential…while for the”.
  25. Line 130. “than these” should read “than those”.
  26. Line 135. “easier” should read “more easily”.
  27. Line 139. “indicates” should read “suggest”,
  28. Line 140. “of a methyl group” should read “of the methyl groups”.
  29. Line 142. “forming C-T” should read “forming a C-T”.
  30. Line 145. “suggests” should read “suggest”.
  31. Line 146. “group in the…velocity of” should read “groups in the…rate of”.
  32. Lines 146,147. I have a suggestion for the oxygen effect, to explain why the rate constant is higher without O2. Toc is a phenol, right? A reasonable first step for its oxidation to quinone likely passes through the formation of a phenoxyl radical (PhO°). That said, considering that with oxygen superoxide (O2°-) can be formed, the following back reaction can explain why the process is slower in air:

PhO° + O2°- + H+ => PhOH + O2, which reforms the initial phenol (Toc in this case).

  1. Line 162. “of <gamma> to” should read “of <gamma> and”.
  2. Line 172. “spectrum and for” should read “spectrum for”.
  3. Line 198. “16 time” should read “16 times”.
  4. Line 200. “on Toc” should read “on the Toc”.
  5. Line 210. “maxima at the” should read “maxima of the”.
  6. Line 211. “Absorption maximum first located” should read “The first absorption maximum was located”.
  7. Line 213. “to either to…or to” should read “to either…or”.
  8. Line 220. “to these” should read “to those”.
  9. Line 231. “hows” should read “shows”.
  10. Line 235. “of each…and its” should read “of all…and the”.
  11. Line 241. “deoxygenized” should read “deoxygenated”.
  12. Line 246. “was analyzed” should read “were analyzed”.
  13. Line 260. “consumed in the” should read “consumed in an”.
  14. Line 276. “of methyl” should read “of the methyl”.
  15. Line 302. Delete “mainly through”.
  16. Line 312. Delete “pending”.
  17. Line 324. Add a full stop “.” after “[40]”.
  18. Line 325. “spectra” should read “spectrum”.

Author Response

Reviewer 2

The author has made a good job while revising the manuscript. Written in better form, even the part I was most sceptical about now looks reasonably convincing. I have still some suggestions for improvement, among which there is also a hint about the possible explanation of the oxygen effect on degradation.

Line 9. “into on the” should read “into the”.

Line 15. “together with and” should read “together with”.

Line 27. “oxidation mainly” should read “oxidation, mainly by”.

Line 29. Delete “presence”.

Line 30. “result” should read “results”.

Line 32. “Oxidation of one electron” should read “One-electron oxidation”.

Line 33. “by second” should read “by a second”.

Line 35. “scavenge a singlet” should read “scavenge singlet”.

Line 37. “fully yet” should read “fully elucidated”.

Line 50. “with UV” should read “with lamps equipped with UV”.

Line 66. “are created” should read “are due to the”.

Line 67. Delete “despite”.

Line 68. “oxidation” should read “abstraction” (also in lines 70 & 72).

Line 70. “that have” should read “that has”.

Line 91. “created for…points in” should read “present for…points are shown in”.

Line 93. “and that” should read “, which”.

Line 99. “these Figures”: which ones? Please specify. “beyond” should read “above” (also in line 104).

Line 100. “form such…check the” should read “form such C-T…check for the”.

Line 101. Delete “transmitted photons”.

Line 103. “higher…UV1” should read “higher compared to the UV1”.

Line 108. “through UG11” should read “through the UG11”.

Line 120. “rate decreased” should read “rate constant decreased”.

Line 121. “reaction for the” should read “reaction for an”.

Line 122. “exponent…when for” should read “exponential…while for the”.

Line 130. “than these” should read “than those”.

Line 135. “easier” should read “more easily”.

Line 139. “indicates” should read “suggest”,

Line 140. “of a methyl group” should read “of the methyl groups”.

Line 142. “forming C-T” should read “forming a C-T”.

Line 145. “suggests” should read “suggest”.

Line 146. “group in the…velocity of” should read “groups in the…rate of”.

Lines 146,147. I have a suggestion for the oxygen effect, to explain why the rate constant is higher without O2. Toc is a phenol, right? A reasonable first step for its oxidation to quinone likely passes through the formation of a phenoxyl radical (PhO°). That said, considering that with oxygen superoxide (O2°-) can be formed, the following back reaction can explain why the process is slower in air:

PhO° + O2°- + H+ => PhOH + O2, which reforms the initial phenol (Toc in this case).

Thank you very much for that clarification. This seems very reasonable, I agree and I will use it in the manuscript: “The first step of the photooxidation reaction is the hydrogen atom abstraction from the hydroxyl group, causing the formation of the tocopheroxyl radical and simultaneously the oxygen superoxide, which, according to the reaction TocO + O2- + H+ => TocOH + O2, partially reforms the initial concentration of the Toc molecule, which may explain the higher reaction rate of argonated samples.

Line 162. “of <gamma> to” should read “of <gamma> and”.

Line 172. “spectrum and for” should read “spectrum for”.

Line 198. “16 time” should read “16 times”.

Line 200. “on Toc” should read “on the Toc”.

Line 210. “maxima at the” should read “maxima of the”.

Line 211. “Absorption maximum first located” should read “The first absorption maximum was located”.

Line 213. “to either to…or to” should read “to either…or”.

Line 220. “to these” should read “to those”.

Line 231. “hows” should read “shows”.

Line 235. “of each…and its” should read “of all…and the”.

Line 241. “deoxygenized” should read “deoxygenated”.

Line 246. “was analyzed” should read “were analyzed”.

Line 260. “consumed in the” should read “consumed in an”.

Line 276. “of methyl” should read “of the methyl”.

Line 302. Delete “mainly through”.

Line 312. Delete “pending”.

Line 324. Add a full stop “.” after “[40]”.

Line 325. “spectra” should read “spectrum”.

Thank you very much again, the number of mistakes again is really very big. All of them were corrected as suggested.

Reviewer 3 Report

This manuscript addresses the photooxidation of a series of tocopherols through absorption and emission spectroscopic studies. The molecules are dissolved in n-hexane (aerated or deoxygenated) and they are irradiated with a polychromatic source. The results show dependence on the substitution pattern in the chromanol ring of tocopherols.

However, the writing and arguments are very difficult to follow. The interpretation of the data is unclear and the conclusions are confusing. These can be attributed not merely to the complexity of the process, but to several basic issues that are overlooked and support my main concern about this paper.

The identity of the hypothetical oxidant species carrying out the transformations observed are not mentioned. If these were oxygen reactive species, how could be the changes observed in the argonated solutions explained? Moreover, direct photochemistry of tocopherols is not discussed. This could lead to photoionization processes which are not analyzed (see for instance: .J. Phys. Chem. A 2011, 115, 29, 8242–8247).

The methodology and purpose behind the analysis of the absorption spectra in S5 and S6 (black lines) is not described. In line 84 and 85, it is suggested that some bands arise from diverse racemic forms. I wonder if racemic forms can show different absorption spectra.

The effect of the UV filters to check the formation of C-T complexes (lines 102 and ss) should be considered more carefully. It is noted that the number of photons transmitted depend also on the spectral distribution of the light source (i.e. the incident irradiation Io(λ)), and not just on the %T of the filters, thus the way to describe the comparison between UG11 and UV1 filters should be revised. Additionally, the reaction rate of a photoinduced reaction depends also on the molar absorption coefficient (ε(λ)) of the reactant and on the quantum yield of the reaction (Φ(λ)). Thus, an overall rate constant should integrate all these factors over the spectral range: Io(λ) (lamp) , %T(λ) (filter), ε(λ) (reactant), Φ(λ) (reaction). Moreover, the occurrence of inner filter effects which may vary with time should be carefully evaluated in photoinduced processes with polychromatic irradiation sources.

Section 2.2. about the rate constants omits any indication on the mechanism assumed and resulting rate law for the estimation of the kinetic parameters, as well as on the data treatment. In this context, is not surprising that a concentration dependence of the rate “constants” arises, since a mechanism which is not validated by the results may have been implicitly considered in order to extract k values.

Lifetimes from time-resolved fluorescence measurements per se are poorly indicative of the identity and dynamics of photoproducts and intermediates. They pose more questions than evidences on the role of the number or position of methyl groups in the chromanol ring in the observed phenomena, and should be complemented with independent determinations.

All these topics must be clarified before considering rather speculative or contradictory explanations such as involvement of dimers or trimers, microviscosity or self-organization effects, and concentration- or wavelength-dependent rate constants.

It seems unavoidable the search of stronger evidences, perhaps from complementary analysis by HPLC or laser flash photolysis experiments, which could bring more light to the subject.

Further minor points that are not clear or need to revision:

Lines 39 and 102:  Chart A1 cannot be found

Line 50: Tocs were irradiated with (through?) UV filters

Line 57: …Their conformation (structure?) differs…

Lines 64, and on: Cuvette thickness (optical path)

Line 187 and ss: EC50 is not defined. If it refers to 50% effective concentration, the sense of using this parameter should be evaluated in the context. The numerical results (499 and 625) have no units.

Caption Figure S4. What´s the meaning of reporting λexc for excitation spectra?

Caption Figures S5 and S6: R2? which is the regression analysis?

Figure S10 a): what are the data shown in back dots and lines?

Author Response

Reviewer 3

This manuscript addresses the photooxidation of a series of tocopherols through absorption and emission spectroscopic studies. The molecules are dissolved in n-hexane (aerated or deoxygenated) and they are irradiated with a polychromatic source. The results show dependence on the substitution pattern in the chromanol ring of tocopherols.

However, the writing and arguments are very difficult to follow. The interpretation of the data is unclear and the conclusions are confusing. These can be attributed not merely to the complexity of the process, but to several basic issues that are overlooked and support my main concern about this paper.

The identity of the hypothetical oxidant species carrying out the transformations observed are not mentioned. If these were oxygen reactive species, how could be the changes observed in the argonated solutions explained? Moreover, direct photochemistry of tocopherols is not discussed. This could lead to photoionization processes which are not analyzed (see for instance: .J. Phys. Chem. A 2011, 115, 29, 8242–8247).

This literature item is very interesting and shows that in the case of, among others, n-hexane (not used in this paper) photooxidation takes place by radical route without the participation of a solvated electron according to TOH + hv => TO• + H•, where TO• is a tocopheroxyl radical. However, in various polar solvents, this radical is formed in the first step of reaction with or without solvated electron. The same results from my experiment, because this radical, having the steady state absorption maximum at 240 nm (Fig. S5b), is formed. The methods used in my experiment do not allow such inference. I would have to use, for example, transient absorption and other set-ups to track the evolution of the system after excitation.

The methodology and purpose behind the analysis of the absorption spectra in S5 and S6 (black lines) is not described. In line 84 and 85, it is suggested that some bands arise from diverse racemic forms. I wonder if racemic forms can show different absorption spectra.

Black lines – Gauss bands after deconvolution. This text was added to captions.
Racemic forms have different spectra in practically whole spectral range from UV to IR, opposite to optically active compounds (enantiomers). I did not study this problem, and I did not find such spectra in literature, but I am almost sure that the two maxima in each first Tocs absorption band are connected with racemic forms.

The effect of the UV filters to check the formation of C-T complexes (lines 102 and ss) should be considered more carefully. It is noted that the number of photons transmitted depend also on the spectral distribution of the light source (i.e. the incident irradiation Io(λ)), and not just on the %T of the filters, thus the way to describe the comparison between UG11 and UV1 filters should be revised. Additionally, the reaction rate of a photoinduced reaction depends also on the molar absorption coefficient (ε(λ)) of the reactant and on the quantum yield of the reaction (Φ(λ)). Thus, an overall rate constant should integrate all these factors over the spectral range: Io(λ) (lamp) , %T(λ) (filter), ε(λ) (reactant), Φ(λ) (reaction). Moreover, the occurrence of inner filter effects which may vary with time should be carefully evaluated in photoinduced processes with polychromatic irradiation sources.

I totally agree, but this is only a qualitative comparison, not a quantitative one. However, such a comparison can be done because both filters transmit photons within the full first alpha-Toc absorption band, so they should produce the same qualitative but obviously not quantitative effects. ε(λ),  Io(λ), (Φ(λ), are practically the same for both filters. In the range beyond alfa-Toc band, practically only the photons transmitted by the UG11 filter, i.e. in the absorption band region of reaction products, work. The only question in this experiment was whether the postulated C-T complexes arose that are not linked by a covalent bond that would be much more difficult to break than the (specific) intermolecular bond. Confirmation of the formation of C-T complexes is not easy and requires a number of experiments, which is beyond the scope of this work.
Moreover, in all cases studied the reactions were pseudo-first order from the photons point of view.

Section 2.2. about the rate constants omits any indication on the mechanism assumed and resulting rate law for the estimation of the kinetic parameters, as well as on the data treatment. In this context, is not surprising that a concentration dependence of the rate “constants” arises, since a mechanism which is not validated by the results may have been implicitly considered in order to extract k values.

In Section 2.2 text related to the rate constants estimated from the absorption spectra was almost completely removed.

Lifetimes from time-resolved fluorescence measurements per se are poorly indicative of the identity and dynamics of photoproducts and intermediates. They pose more questions than evidences on the role of the number or position of methyl groups in the chromanol ring in the observed phenomena, and should be complemented with independent determinations.

The lifetimes, and TRES in particular, clearly indicate that the emerging long lifetime of approx. 6 ns is associated with the newly formed fluorescence band with a maximum of approx. 298 nm. This band is evident in the case of gamma- and delta-Toc. However, in the case of other tocopherols, I did not observe this band, because it probably coincides with the alpha and beta-Toc main band and has a lower intensity (quantum yield). Moreover, in the case of gamma and delta, an intermediate product with a lifetime of approx. 3 ns appears. Since the above-mentioned tocopherols differ in the presence of one methyl group, it seems that it has a decisive influence on the formation of this new band in the range below 300 nm. It is obvious that it is difficult to say anything about the formation mechanism on the basis of the presented results, but the results seem to form grounds for further research.

It seems unavoidable the search of stronger evidences, perhaps from complementary analysis by HPLC or laser flash photolysis experiments, which could bring more light to the subject.

I totally agree, but it needs to be done in further research also using for example GC-MS or NMR. This work is only a signal of some interesting and strange phenomena occurring as a result of tocopherols exposure to irradiation, and I hope it will stimulate other researchers with the right research kit to explain the observed effects.

Further minor points that are not clear or need to revision:

Lines 39 and 102:  Chart A1 cannot be found

Line 50: Tocs were irradiated with (through?) UV filters

Line 57: …Their conformation (structure?) differs…

Lines 64, and on: Cuvette thickness (optical path)

Line 187 and ss: EC50 is not defined. If it refers to 50% effective concentration, the sense of using this parameter should be evaluated in the context. The numerical results (499 and 625) have no units.

Caption Figure S4. What´s the meaning of reporting λexc for excitation spectra?

Caption Figures S5 and S6: R2? which is the regression analysis?

R2 is connected with Gauss decomposition procedure and means the quality of fitting. 

Figure S10 a): what are the data shown in back dots and lines?

All of the mistakes and errors were corrected as suggested.

Round 2

Reviewer 3 Report

The manuscript shows some improvements. Most of the comments and questions posed in the first review have been answered. However, I must insist in the observations that have not been addressed and deserve the effort to be clarified before publishing the article. 

Section 2.2 still lacks an explanation about how the rate constants were estimated from experimental data (data treatment). Should they come from the mono-exponential fits of absorbance-time profiles, then this should be clearly stated in the main text (not just in caption to figures). I agree that the data is difficult to interpret due to the overlap of bands, so I suggest to condense this section and just keep the relevant findings.

Lines 148 & 149: This paragraph is very confusing. It is not obvious why the concentration of O2 could be higher than that of tocopherol´s even after deoxygenation, nor the connection of this with the effect of microviscosity.

Lines 186 and ss: EC50 is not defined. The parameter EC50 is usually applied to dose-response relationships, which is not this case.

Figure S10: Please explain more clearly what are the data shown in black dots and lines in panel a). What do the author mean by “red and blue” range? .

Further edition is required in the following items:

Lines 78, 80: approximately instead of “approx.”

Line 80: A connector seems to be missing in the phrase:  “…and its spectrum does not overlap with that of Tocs, (?) no influence on the left isosbestic point..”

Chart S1. Molecular structures of Tocopherols

Author Response

Reviewer 2

The manuscript shows some improvements. Most of the comments and questions posed in the first review have been answered. However, I must insist in the observations that have not been addressed and deserve the effort to be clarified before publishing the article.

 Section 2.2 still lacks an explanation about how the rate constants were estimated from experimental data (data treatment). Should they come from the mono-exponential fits of absorbance-time profiles, then this should be clearly stated in the main text (not just in caption to figures). I agree that the data is difficult to interpret due to the overlap of bands, so I suggest to condense this section and just keep the relevant findings.

Section 2.2 has been modified. New text marked in yellow was added: “Reaction rate constants were calculated for the region 280 – 300 nm (Tocs) using one exponential decay function (Abs (t) = Span*exp(-K*t) + Plateau), and for the regions: 240 – 270 nm (products) and 320 – 400 nm (C-T complexes) using one exponential association function (Abs (t) = Abs(0) + (Plateau-Abs(0))*(1-exp(-K*t))) with 1 nm step starting with the initial spectrum (when it was possible) and ending with one of the last ones.”

Lines 148 & 149: This paragraph is very confusing. It is not obvious why the concentration of O2 could be higher than that of tocopherol´s even after deoxygenation, nor the connection of this with the effect of microviscosity.

This sentence has been removed.

Lines 186 and ss: EC50 is not defined. The parameter EC50 is usually applied to dose-response relationships, which is not this case.

Text was changed (line 188-189): ” It can be seen that the rate growth is higher for the argonated sample, which confirms the higher reaction rate of this sample.”

However, there are works that employi equations using ligand-protein interactions to describe a time dependent reaction (Daniel Mann, Udo Höweler, Carsten Kötting, and Klaus Gerwert. Elucidation of Single Hydrogen Bonds in GTPases via Experimental and Theoretical Infrared Spectroscopy. Biophysical Journal 112, 66–77, January 10, 2017.)

Figure S10: Please explain more clearly what are the data shown in black dots and lines in panel a). What do the author mean by “red and blue” range? .

It has been corrected: ”Figure S10. a) Time-course of absorbance of δ-Toc in the range from 294 to 306 nm, black dots – absorbances for a given wavelength and time. Red and blue dots and lines show changes at absorption maxima. b) Values of the rate constants in above range. Sample was non-argonated. Black dots – rate constants. Bars = SD.”

Further edition is required in the following items:

Lines 78, 80: approximately instead of “approx.”

It has been corrected.

Line 80: A connector seems to be missing in the phrase:  “…and its spectrum does not overlap with that of Tocs, (?) no influence on the left isosbestic point..”

It has been corrected.

Chart S1. Molecular structures of Tocopherols

It has been corrected.

This manuscript is a resubmission of an earlier submission. The following is a list of the peer review reports and author responses from that submission.

Round 1

Reviewer 1 Report

REVIEW OF THE MANUSCRIPT "PHOTOOXIDATION OF TOCOPHEROLS", BY B. SMYK

This paper is potentially interesting, but it is also poorly written and poorly organised. Interpretation of the results is also doubtful, as very complex explanations are provided when much simpler data interpretation looks much more likely.

Really, I cannot recommend acceptance in the present form. My recommendation for improvement are provided below. In my opinion, the suggested changes (including the interpretation of the results) are critical for the final decision on this manuscript.

GENERAL AND SPECIFIC REMARKS

a) Lines 50,51. UV1 filter, both filters. Because the Methods section is at the end, these statements are incomprehensible here. Please provide the minimum information for the readers to understand what are these filters. Same issue in lines 102,103: specify what the UG11 and UV1 filters are.

b) Lines 56-58. "Such changes…(Chart A1)". This sentence looks terribly speculative. My advice is to be more cautious, saying something as: "We can see a different behaviour, probably because of different structures, and the Toc structures mainly differ for…".

c) Lines 58-72. This part is quite messy. "suggest the" should read "suggest an".

"or two molecules. … two electron oxidations. There are many" should read "or two molecules, which might be different oxidized and non-oxidized Toc form(s). There are many.".

"reports of such forms" should read "reports on the oxidized forms".

"systems. However, it seems" should read "systems, and it seems".

"the main oxidized form" should read "the main one".

"having the maximum…at approximately 260" should read "that has an absorption maximum at approximately 260".

"which is formed" should read "and is formed".

"with the maximum [33]" should read "with a maximum [33]".

d) Lines 100 and 105-112. Here I do not understand. The experiments are not such as to suitably differentiate among the CT complexes TQ-Toc and TQ-FP (or Toc-FP). Therefore, all this discussion looks deprived of much foundation.

e) I am very sorry, but the whole section 2.2 is nonsensical. Please, throw it away. The effect of O2 is not something short of miraculous: it is the expected trends if excited states like the singlet and the triplet ones are involved in photoreactivity. Very simply, O2 can physically quench the excited singlet states. Moreover, be Toc the ground state molecule and 3Toc* the excited triplet state. The effect of O2 on photodegradation is quite simply explained by the following process (plus singlet-state quenching): (ISC = inter-system crossing)

Toc + hv =(ISC)=> 3Toc*

3Toc* = = = = > Reaction products

3Toc* + O2 => Toc + O2

3Toc* + O2 => Toc + 1O2

Suggestion: totally delete section 2.2, cut figure 1b and replace it with a figure similar to 1a, but reporting the corresponding spectral evolution without O2. You say that the reaction is faster in Ar, and that's all.

Moreover, finding decreasing k with increasing concentration is much more simply exlained by saturation of absorption, but the way k was calculated in section 2.2 is quite doubtful and might well be wrong (I would bet it is, although the calculation procedure is not clear at all). At this level, it would be much better to carry out a qualitative explanation of the results.

f) Lines 210-212. "Its shape…reconstructed". This statement is far from being straightforward or obvious. A reference is needed here to support it.

g) Lines 218-222. Quite obscure/awkward. Report the calculation procedure (used formulae) in the Methods section, it will be clearer.

h) Line 233. What is a "level of deoxygenatiuon"? Do you mean the concentration of dissolved oxygen? Other stuff? Please, be clearer.

i) Line 239-240. "It is also…for all molecules". Quite incomprehensible. Please rephrase for clarity. Probably, this statement (when made clear) will also need a reference in support.

j) Line 241. "photooxidation" in Ar atmosphere, without oxygen, is not the first thing that comes to mind as likely. Please explain better.

k) Line 270. "their spectra were not shown". To avoid redundance, or because they were difficult to measure (e.g., too low intensity)? It is a potentially interesting piece of information, please explain better.

l) Line 293. Spell out the "TRES" acronym.

m) Conclusions are to be deleted completely. I do not think that the results are difficult to explain, after all. I could make two very simple proposals:

1) Effect of concentration: saturation of absorbance. When all incoming radiation is absorbed, the solution cannot absorb more even if the concentration is increased, thus photodegradation becomes less efficient and the value of k gets lower with increasing concentration. The percentage of incoming radiation absorbed by the solution can also be quantitatively measured to make a check on this statement.

2) Effect of O2: scavenging/quenching of the excited states. The quenching of fluorescence is demonstrated experimentally, triplet-state quenching by O2 is commonplace in photochemistry and quite expected, even in the absence of dedicated measurements.

Therefore, please rewrite the conclusions.

MINOR ISSUES

  1. Line 8. "of the investigation on" should read "of an investigation into".
  2. Line 13. "in the argonated" should read "in argonated".
  3. Line 14. "with the…was formed and" should read "with a…appeared together with".
  4. Line 15. Delete "were appeared".
  5. Line 24. "as a very important vitamin" should read "as very important analogs of".
  6. Line 26. "the bound" should read "bound".
  7. Line 27. "before oxidation…scavenge" should read "from oxidation, … scavenging".
  8. Line 28. "reaction…of -OH group presence" should read "reactions…of the occurrence of a -OH group".
  9. Line 30. "result…prolong" should read "results…propagate".
  10. Line 31. "form dimer…the planar" should read "form a dimer…a planar".
  11. Line 32. "complexes…Oxidation of one electron can" should read "oligomers…One-electron oxidation can".
  12. Line 35. "a singlet oxygen in" should read "singlet oxygen by".
  13. Line 36. "21] but" should read "21] which, "
  14. Line 37. "this had not been" should read "however, had not been fully". "in my previous…moreover" should read "yet. Moreover,".
  15. Line 38. "than the other" should read "of all the".
  16. Line 42. Delete "ultraviolet".
  17. Line 52. "Absorption" should read "The absorption".
  18. Line 53. "these" should read "those" (thrice). Similar issue in lines 122,123, 207(twice).
  19. Line 55. "then gap time increased" should read "which continued".
  20. Line 58. "are created…suggest the" should read "were formed…suggest an".
  21. Line 72. "To find" should read "To find the".
  22. Line 76. Delete "spectrum". "seems that" should read "seems that the".
  23. Line 77. "at the maximum" should read "with maximum".
  24. Line 82. "with the maxima" should read "with maxima" (also in line 83).
  25. Line 86. "The one with the" should read "The band with".
  26. Line 89. "and peak" should read "and the peak".
  27. Lines 92,93. "the second…and were created" should read "a second…that might be".
  28. Line 97. "allow forming" should read "allow for forming".
  29. Line 107. "at maximum" should read "at the absorption maximum".
  30. Line 111. "from the range" should read "in the range".
  31. Line 115. "for regions" should read "for the regions".
  32. Lines 119,120. "for presentation" should read "to estimate the". "at the maximum and of" should read "at its absorption maximum and those of".
  33. Line 121. "complexes" should read "complexes at".
  34. Line 123. "and these" should read "and of those".
  35. Line 127. "285" should read "285 nm".
  36. Line 144. "FP and" should read "FP and the".
  37. Line 147. "slower than" should read "slower than the".
  38. Line 149. "because were" should read "because they were".
  39. Lines 153,154. "-4" should be an exponent.
  40. Line 155. "separate" should read "separately".
  41. Line 157. Delete "range".
  42. Line 158. "is the first" should read "is a first".
  43. Line 159. "constant to" should read "constant up to".
  44. Line 178. "because of not" should read "because the reaction would not be".
  45. Line 179. Delete "reaction".
  46. Line 193. "10th" should read "10".
  47. Line 207. "to these of <gamma>- to" should read ", those of <gamma> were similar to".
  48. Line 208. Delete "the presented". "parallel to" should read "parallel for".
  49. Line 211. "the changes" should read "changes".
  50. Line 212. "correspond to the" should read "correspond to a".
  51. Line 216. "because…excepts" should read "because the fluorescence spectra are homogeneous except".
  52. Line 217. "overlaps" should read "overlap".
  53. Line 220. "the spectrum…to the same" should read "the fluorescence spectrum…at the same".
  54. Line 221. "area" should read "the area".
  55. Line 222. "overlapping spectra…of excitation" should read "spectral overlap…of the excitation".
  56. Line 228. "influences only on the" should read "affects only the".
  57. Line 229. "and dividing by…at excitation" should read "and the division by…at the excitation".
  58. Line 230. "give relative" should read "give the relative".
  59. Line 237. "for particular" should read "it holds for any".
  60. Line 241. Delete "As it was".
  61. Line 247. "two times higher" should read "twice wider".
  62. Line 249. "of the excitation" should read "of excitation".
  63. Line 258. "at excitation" should read "at the excitation".
  64. Line 260. "by maxima at excitation" should read "by the maxima of the excitation".
  65. Line 261. "Absorption…one located" should read "The first absorption…was located".
  66. Line 261. "one at" should read "occurred at". "presented" should read "shown".
  67. Line 263. "to nor to…neither to" should read "to either…or" (never use double negative).
  68. Line 264. "band product" should read "band of a product".
  69. Line 267. "at excitation…at 265" should read "at an excitation…of 265".
  70. Line 269. "such a" should read "as such a".
  71. Lines 272,273. "for one excitation wavelength…and one observation wavelength" should read "for excitation…and emission".
  72. Line 280. "Figure…two-exponential" should read "Figures…bi-exponential".
  73. Line 281. "three-exponential" should read "tri-exponential".
  74. Line 284. "are "visible"" should read "could be seen".
  75. Line 288. "very small…was imposed…are longer" should read "very low…was interfered…were longer".
  76. Line 292. "than the short lifetimes are" should read "if lifetimes are longer".
  77. Line 295. "<beta>- and <gamma>-" should read "<beta>-, as well as <gamma>-". "of long" should read "with long".
  78. Line 300. "by spectrum" should read "by the spectrum".
  79. Line 302. "and long" should read "and the long".

Reviewer 2 Report

In the present work, this manuscript introduces the research results of photooxidation of Tocs dissolved in argonated and non-argonated n-hexane. During the irradiation, steady-state absorption and fluorescence spectra and lifetime were measured. The results obtained show that the rate constant is concentration-dependent, and when the concentration increases, the rate constant decreases. In addition, the following results are obtained. For all Tocs, the reaction rate is higher in argon solution. During the irradiation of γ- and δ-Toc, a new emission band was formed, with a maximum emission wavelength of 298 nm, and absorption and fluorescence bands connected to the charge transfer (C-T) complex appeared. If the author can solve all the following problems, we will suggest publishing this article on Molecules.

  1. The title is not precise enough and too general. A more detailed title will help readers quickly understand this manuscript.
  2. There is too much content in each paragraph, and a paragraph should be divided into several small paragraphs according to the content. Excessive content in each paragraph makes the regulations unclear and inconvenient for readers to read.
  3. The pictures in the supporting information should not be represented by the letter “A”. It is easy to be confused with the letter “a” in the pictures in the manuscript. The letter “S” is recommended. And in line 129, the mention of "Figure A9a, S9c" is obviously wrong.
  4. Figure 2 is mentioned in the manuscript, but I did not find it.
  5. Regarding the formation of C-T complexes, it is not convincing enough. Can it be explained by transient spectroscopy? You can refer to Angew. Chem. Int. Ed. 2019, 58, 11642-11646.
  6. It is not rigorous to judge the fluorescence quantum yield by comparing the intensity of absorption and emission in the manuscript. The intensity of absorption and emission will be interfered by factors such as concentration. It is recommended to perform a fluorescence quantum yield test to illustrate the magnitude of the fluorescence quantum yield of several molecules.

Reviewer 3 Report

The paper presents the kinetic study of the photooxidation of tocopherols (Tocs) in argonated and non-argonated n-hexane by monitoring the temporal evolution of their UV and fluorescence spectra. Investigating the photostability and the photoprotective effect of these biomolecules is crucial due to the many reasons also listed by the introduction of the manuscript, therefore the word could be potentially interesting to the scientific community. However, a fundamental error in the data analysis sadly makes all the results and conclusions questionable and therefore the manuscript cannot be recommended for acceptance in its present form.

What I mean is that the kinetic profiles (therefore the reaction rates) were simply derived from the absorption maxima and at arbitrarily chosen (!) wavelengths (if I get it right), therefore they cannot be trusted due to the extensive overlapping of the broad UV bands. I suspect that the "plateau" seen in Figure A12a in the beginning of the kinetic profiles is caused by the overlapping of the bands. More precisely, in the case of d-Toc, a new very broad band arises at 278 nm (based on the deconvoluted spectrum in Figure A7b), which could easily offset the decrease of the band at 297 nm in the early stages of the photooxidation. This is what should cause the plateau-like structure in the kinetic profile in Figure A12a.

Furthermore, one cannot simply fit a single exponential decay on a kinetic curve that looks like the ones in Figure A12a. Rate constants derived from these fits cannot be trusted, making the conclusions drawn from them and discussions about them unfortunately completely irrelevant. One telltale sign is  "concentration dependence" of the rate constants often cited by the manuscript. If a rate constant (a constant!) is dependent on the concentration, then there must be something going on either with the kinetic model or with the data analysis. I suppose, in this case, this apparent dependence on concentration is due to the extensive overlapping of the broad UV bands. Note that the more concentrated the solution is, the more broad and overlapping the UV bands are, and they will inevitably interfere more with each other, which is evidently seen in Figure A12a.

What I would suggest is that the author should try and use the integrated area of the Gaussians in the deconvoluted spectra presented in Figures A6 and A7 instead and plot them versus reaction time to create the kinetic curve. This should eliminate the problem of overlap, and I would suspect that the kinetic profiles obtained this way would not exhibit such plateaus in the initial stage of the reaction. Once the this major problem is tackled, I would like to encourage author to resubmit the manuscript with the new findings and their discussion.

Other, minor remarks:

1. Citations are pretty much useless in the current form, numbers are used in the main text and the references are in alphabetical order at the end of the manuscript. Hence the references simply cannot be checked.

2. Figure 2 is missing. Does the author mean Figure 1b whenever he refers to Figure 2?

Reviewer 4 Report

Professor Bogdan Smyk investigated the photooxidation of tocopherols by steady-state and time-resolved spectroscopic studies. The author suggested oxidation species and kinetics based on experimental results and simulation fitting. I think this work will provide useful scientific insights on the photoproducts of tocopherols. Therefore, I recommend this paper for publication in Molecules after minor correction.

Comments:

- The author should remove the sentence “ Further investigations are, however, needed to gain more in-depth knowledge of the observed phenomena” from the abstract and conclusion sections, last lines:

- Chart A1 maybe include in the main text.

- To keep consistency, inset label in figure (1b), Figure 3c and d, Figure A3 and Figure A4 may rearrange in order of α, β, γ, and δ.

- Figure 3: b → b).

- The Author should provide the monitoring wavelength for the excitation spectra in the figure caption (in Figure 4, Figure A5).

- There is no explains in the caption and main text for Figure 1b. Especially, 1.35 × 10-4 ÷ 2.70 × 10-4 M, etc., are not explained. Why did not show only the calculation results?